# Diel Variations of Extracellular Microcystin Influence the Subcellular Dynamics of RubisCO in *Microcystis aeruginosa* PCC 7806

**DOI:** 10.3390/microorganisms9061265

**Published:** 2021-06-10

**Authors:** Arthur Guljamow, Tino Barchewitz, Rebecca Große, Stefan Timm, Martin Hagemann, Elke Dittmann

**Affiliations:** 1Department of Microbiology, Institute for Biochemistry and Biology, University of Potsdam, Karl-Liebknecht-Street 24/25, 14476 Potsdam-Golm, Germany; barchewitz@uni-potsdam.de (T.B.); rgrosse@uni-potsdam.de (R.G.); editt@uni-potsdam.de (E.D.); 2Department of Plant Physiology, Institute of Biological Sciences, University of Rostock, Albert-Einstein-Street 3, 18059 Rostock, Germany; stefan.timm@uni-rostock.de (S.T.); martin.hagemann@uni-rostock.de (M.H.)

**Keywords:** cyanobacterial bloom, *Microcystis*, microcystin, RubisCO, extracellular signaling

## Abstract

The ubiquitous freshwater cyanobacterium *Microcystis* is remarkably successful, showing a high tolerance against fluctuations in environmental conditions. It frequently forms dense blooms which can accumulate significant amounts of the hepatotoxin microcystin, which plays an extracellular role as an infochemical but also acts intracellularly by interacting with proteins of the carbon metabolism, notably with the CO_2_ fixing enzyme RubisCO. Here we demonstrate a direct link between external microcystin and its intracellular targets. Monitoring liquid cultures of *Microcystis* in a diel experiment revealed fluctuations in the extracellular microcystin content that correlate with an increase in the binding of microcystin to intracellular proteins. Concomitantly, reversible relocation of RubisCO from the cytoplasm to the cell’s periphery was observed. These variations in RubisCO localization were especially pronounced with cultures grown at higher cell densities. We replicated these effects by adding microcystin externally to cultures grown under continuous light. Thus, we propose that microcystin may be part of a fast response to conditions of high light and low carbon that contribute to the metabolic flexibility and the success of *Microcystis* in the field.

## 1. Introduction

Cyanobacterial strains of the genus *Microcystis* are infamous for the formation of freshwater blooms, which pose a threat to humans and animals due to the production of the potent hepatotoxin microcystin (MC) [1]. Climate change and increasing eutrophication in lakes further promote the mass development of these cyanobacteria that leads to an ecological collapse in affected freshwater habitats connected with a considerable economic loss [2,3]. The cyclic heptapeptide MC is produced by a giant non-ribosomal peptide synthetase (NRPS) assembly line linked with high metabolic costs for the producing cells [4]. The elaborate biosynthetic machinery is evolutionarily old and was already present in the last common ancestor of all recent cyanobacteria and, hence, prior to the evolution of eukaryotic predators [5]. While the ability to produce the cyanotoxin and other non-ribosomal peptides was lost in many cyanobacterial phyla it was retained in phylogenetically distant bloom-forming species suggesting a close connection to the bloom-forming lifestyle [5]. Yet, *Microcystis* blooms are composed of different *Microcystis* chemotypes that produce varying cocktails of non-ribosomal peptides where an individual strain is typically able to synthesize 2–4 different types of non-ribosomal peptides [6]. Signature peptides of *Microcystis* spp. include MCs, cyanopeptolins (CP), aeruginosins (AER), microginins (MG), anabaenopeptins (AP), and aeruginoguanidines (AG) [6,7]. Hence, only a fraction of strains produces MC as part of their chemical repertoire, while other strains produce other combinations of non-ribosomal peptides.

Loss of MC in the *mcyB* mutant of the laboratory strain *M. aeruginosa* PCC 7806 leads to a severe phenotype including differences in cell size, pigmentation, and inorganic carbon adaptation [8,9,10,11]. Existing studies on the function of MC for the producing cells can be principally subdivided into intracellular and extracellular functional analyses [12]. MC is primarily accumulated inside the cells. At high cell densities, MC can form SDS-stable conjugates with proteins, particularly with enzymes of the Calvin-Benson-Bassham (CBB) cycle that form high-molecular mass protein complexes in cells of *Microcystis* [13,14,15]. MC is then found in distinct spots in the cytoplasm of *Microcystis* cells [13,16]. MC-binding to proteins is also stimulated at high light conditions and under oxidative stress. A predominant binding partner of MC is the key enzyme of the CBB cycle, ribulose-1,5-bisphosphate carboxylase/oxygenase (RubisCO). Coincidently, RubisCO assembly, dynamics, and subcellular localization deviate from the textbook view on cyanobacterial RubisCO and the carbon-concentrating mechanism (CCM) [13,16]. Specifically, the two subunits of RubisCO, RbcL, and RbcS are conditionally separate in *Microcystis,* and RubisCO is frequently associated with the cell membrane rather than located in carboxysomes [13,16]. Metabolomic experiments have revealed pronounced differences in the accumulation of products of the RubisCO oxygenation reaction in *Microcystis* PCC 7806 and the model cyanobacterium *Synechocystis* PCC 6803 under high light conditions that may correspond to differences in the subcellular localization of RubisCO [17]. It is noteworthy that differences in the CCM of *Microcystis* go beyond observed peculiarities at the level of RubisCO. Individual strains of *Microcystis* also lack specific bicarbonate transporters and show strong variations in the adaptation to either limiting or elevated concentrations of inorganic carbon [18,19]. Consequently, blooms of *Microcystis* show enormous phenotypic plasticity with regard to the inorganic carbon adaptation which makes them resilient at the community level to varying concentrations of CO_2_ in the atmosphere including the predicted future climate change [18].

While MC is principally located inside cells, small amounts of the peptide are released. The MC biosynthesis operon encodes the ABC transporter McyH that may be responsible for the secretion of MC, albeit the mechanism of the MC transport is currently unknown [20]. The level of MC in the medium is positively correlated with cell density [14] and light intensity [21]. Extracellular MC, in turn, was found to stimulate the aggregation of *Microcystis* cells and may play a role in maintaining colony formation of *Microcystis* [22]. A few studies have suggested that MC functions as an infochemical or a quorum sensing (QS)-like metabolite. MC was found to stimulate the accumulation of its biosynthetic proteins, thus fulfilling a key characteristic of QS systems, the autoinduction [23]. Yet, a transcriptomic analysis of the impact of externally added MC suggested a limited cellular response at the level of gene expression. This experiment, however, was performed under low light conditions (16 µE m^−2^ s^−1^) where only a cryptic secondary metabolite gene cluster showed a strong response to MC addition [24]. While several studies thus point towards a role of MC as signal and a cross-talk between different secondary metabolite gene clusters, extracellular contributions to the functional role of MC are not well understood.

Here, we have analyzed the dynamics of extracellular MC and other secondary metabolites in the model strain *M. aeruginosa* PCC 7806 during a diel cycle at two different cell densities. Comparison of the secretion dynamics with the dynamics of RubisCO showed close correlations suggesting a possible role of extracellular MC in the condensation and localization of RubisCO. This hypothesis was tested using MC addition experiments which point to a regulatory function for secreted MC, linking both its intracellular and extracellular role to the control of inorganic carbon fixation in *Microcystis*.

## 2. Results

### 2.1. Extracellular Dynamics of Secondary Metabolites in Microcystis

Given that previous studies reported a major impact of the cellular density of *Microcystis* cultures on both the secretion of MC and the binding of MC to proteins, we cultivated liquid cultures of *Microcystis* at two different cell densities with a 16 h:8 h light:dark photoperiod. Light was adjusted to 55 µmol photons m^−2^ s^−1^ during the light phase. Cultures were entrained with the diel rhythm at an OD_750_ of 0.1 and the experiments were started at OD_750_ 0.25 (low cell density) or 0.5 (high cell density), respectively. Both experiments were pursued for 28 h. In the low-density experiment, cultures grew to an OD_750_ of 0.5 in the course of the experiment while in the high-density experiment cultures grew to a maximum OD_750_ of 0.65, indicating a limiting effect on growth due to shadowing (Appendix A). Cell pellets and supernatants of all biological replicates were analyzed for the accumulation of potentially bioactive peptides by high performance liquid chromatography (HPLC). MC and CP were identified by HPLC validation runs including analytical standards. Cells of the low-density experiment showed an accumulation of MC with its two isoforms MC-LR and [D-^3^Asp]-MC-LR and cyanopeptolin 963A (CP 963) among other unidentified peaks with no considerable variations during the light and dark periods. No significant amounts of the non-ribosomal peptides were detected in the supernatant throughout the experiment (Figure 1B). MCs and CPs were also dominant in cell pellets of high-density cultures (Figure 1D). In contrast to the low-density experiment, substantial amounts of MCs, CP and additional unidentified compounds were released to the medium (Figure 1E). Strikingly, extracellular peaks showed pronounced fluctuations which differed between individual substances, some of which could only be detected in individual samples. Secretion pattern were not the same during the first and second light period of the experiment suggesting that secretion of peptides does not solely depend on the light-dark cycle. MC-LR, [D-Asp^3^]-MC-LR and CP 963 amounts were quantified based on analytical standards (Figure 2). MC-LR was the major MC isoform and showed a slight decline in the cell pellets in the course of the diel experiment. The second isoform [D-Asp^3^]-MC-LR was following a similar trend albeit at a lower concentration (Figure 2A). In the supernatant, MC-LR showed a sharp increase at the onset of darkness after 20 h of the experiment and remained high until the end of the experiment (Figure 2B). This extracellular increase during the night corresponded to an intracellular decrease. Again, [D-Asp^3^]-MC-LR showed a similar trend at a lower concentration level. CP 963 showed only marginal variations in the cell-bound fraction but displayed strong changes in the supernatant fraction (Figure 2C,D). In all biological replicates, the CP 963 peptide was only released after transition from light to dark between 16 h and 24 h of the experiment (Figure 2D).

### 2.2. Diel Dynamics of Proteins and Protein-Bound MC

From cells of each sample, proteins were isolated for all biological replicates and analyzed using SDS-PAGE. The amount of the subunits of RubisCO, RbcL and RbcS, the carboxysome shell protein CcmK, and protein-bound MC was quantified using specific antibodies in Western-Blot experiments. RbcS and CcmK did not show considerable variations throughout the experiment (Figure 3A,B). The amount of RbcL, however, displayed a strong decline at the end of the experiment after 26 h, which was even more pronounced after 28 h, at the beginning of the second light phase. These data agree with previous observations that the turnover of the large and the small subunit of RubisCO differs in *Microcystis* where the small subunit is generally more stable than the large subunit. MC-binding to proteins strongly increased at the end of the first light phase after 16 h but strongly declined again during darkness until 24 h. Both the decline of RbcL and protein-bound MC became visible (Figure 3B) when high extracellular levels of MC were observed (Figure 2B). We further tested variations in the subcellular localization of RbcL and CcmK during the experiment using immunofluorescence microscopy (IFM). In agreement with the immunoblotting experiments, no major variations were observed, the CcmK-derived fluorescence signals were visible in the cytoplasm of cells and formed characteristic shapes previously reported as carboxysomal shells [13]. At the beginning of the experiment, RbcL was mostly detected in the cytoplasm and signals were partially overlapping with CcmK signals indicating its accumulation inside carboxysomes. At the end of the light phase, however, the cytoplasmic proportion of RbcL declined and more RbcL was detected underneath the cell membrane as previously reported for high-light treated cells [13]. At these time points, RbcL signals were not much overlapping with CcmK signals (Figure 3C). In the final phase of the experiments, the overall amount of RbcL decreased both in the cytoplasmic and underneath the membrane thereby corresponding to the low amount detected in the immunoblotting experiment.

To evaluate whether the decrease in the amount of RubisCO corresponded to a decrease in the amount of the RubisCO products 3-phosphoplycerate (3-PGA) and 2-phosphoglycolate (2-PG), both metabolites were quantified along the time course in each of the replicates. Intracellular steady-state levels of 3-PGA slightly decreased during night. Virtually no 3-PGA was released to the medium (Figure 4). Only low intracellular levels of 2-PG could be detected. Intracellular amounts of 2-PG reached a maximum at the beginning of both light phases. In contrast to 3-PGA, almost equal amounts of 2-PG were found in the cells and released into the medium (Figure 4).

### 2.3. Extracellular MC Triggers Dynamics of RubisCO

As the diel experiments revealed a co-occurrence of low levels of free intracellular MC, an increase in MC-protein binding and high amounts of extracellular MC, we tested the hypothesis that high amounts of extracellular MC may trigger the binding of MC to proteins. We conducted an experiment where MC was added to low-light adapted cultures of *M. aeruginosa* PCC 7806 of two different cell densities that were then subjected to high-light irradiation for two hours. Aliquots of the culture were treated with 0 or 100 ng/mL MC-LR, respectively, and the increase of MC in the growth medium was confirmed by HPLC (Appendix A). Total RNA was isolated to follow the transcription by qRT-PCR of *rbcL*, *rbcS*, *mcyA* and a biosynthesis gene of a cryptic PKSI/PKSIII gene cluster that was found earlier to show a strong transcriptional reaction to externally applied MC [24]. Although in most cases transcript amounts were slightly higher when MC was applied externally, the log_2_-fold change values did not exceed 1.1 (Appendix A). Additionally, total cellular proteins were isolated and the amounts of RbcL, RbcS and of protein-bound MC were evaluated by immunoblotting analysis. While addition of MC had only a negligible effect on the amount of RbcL, RbcS showed a slight decrease in high cell density cultures (Figure 5B). In contrast to that, the fraction of MC-bound proteins clearly increased in the presence of externally applied MC. This effect was more pronounced with cultures of higher cell densities, although it was also perceptible in low density cultures. Aliquots of the same culture were also analyzed using immunofluorescence microscopy (Figure 5C). Under control conditions, both RbcL and RbcS showed a clear cytoplasmic localization. While RbcL was largely evenly distributed throughout the cytoplasm, RbcS showed a characteristic spot-like accumulation reminiscent of carboxysomes in size, number and distribution. Under high light conditions, both RbcL and RbcS showed a beginning relocation to the cell’s periphery but both continued to be present in the cytoplasm. With additional MC present in the growth medium, however, both proteins showed a clear accumulation in distinct spots underneath the cell membrane.

## 3. Discussion

Due to the grave ecological and economic consequences mass occurrences of the potent toxin MC can have, it is among the most well-studied bacterial secondary metabolites. Despite that, its actual biological function is still largely unclear and has only begun to emerge in recent years. After it was dismissed as having evolved as a sole feeding deterrent [5], evidence has accumulated showing MC to be involved in a number of biological processes, both intra- and extracellularly. In accordance with the considerable amounts of MC that are frequently found to be released as a free, unbound molecule, there is evidence supporting its role as an infochemical that maintains an intricate system of cell-cell interaction [23]. Doubtlessly, however, microcystin also plays an important intracellular role that seems to be rooted in the central carbon metabolism. Not only does it bind to a number of key enzymes of the CBB cycle, including RubisCO [14,25], its influence becomes tangible in key metabolites of carbon fixation pathways that accumulate much differently in the wild type and its microcystin-free mutant [17]. Consequently, the presence or absence of microcystin correlates with pronounced differences in the ability to cope with fluctuations in carbon availability [11]. In earlier experiments we have proposed a mode of action for intracellular MC as its ability to bind proteins transiently and non-covalently plays a role in determining the formation and subcellular localization of RubisCO and other CBB enzymes into protein super complexes of carbon fixation [13].

Here, we provide first evidence that indicate a direct link between external MC and its intracellular role in reshaping the carbon fixation machinery. Our diel experiment shows a correlation of the amount of extracellular MC with the phases of a light-dark cycle, showing low levels and a slight decrease of MC during the light phase that turns into a steep MC increase of more than 3-fold immediately following the onset of the dark phase. Interestingly, the extracellular accumulation of MC during darkness coincides with a steady decrease of its free intracellular portion and a concomitant quick increase in the binding of MC to proteins that spans the transition phase from light to dark. This in turn correlates with a relocation of RubisCO that is predominantly cytoplasmic, presumably in carboxysomes, at the beginning of the light phase but becomes at least partly accumulated in distinct regions underneath the cell membrane towards the beginning of the night. While we know from previous experiments that this relocation and condensation into distinct CBB enzyme complexes is linked to a shift from lower to higher light intensities [13], our experiment here clearly shows that this effect can be boosted by external MC either released from cells in the diel cycle or added into the surrounding medium. In our immunofluorescence micrographs, for instance, we see a trend of high light induced relocation and peripheral accumulation of RubisCO without the addition of external MC. This process, however, is more pronounced and more “complete” when MC is added to the growth medium. In both our experiments there is a clear correlation between an elevated amount of microcystinilated intracellular proteins and the relocation of RubisCO to the cell membrane. In the light-dark experiment, increased MC binding to proteins precedes the increase in extracellular MC, while in our MC addition experiment the former seems to be induced by the latter. This can perhaps be explained by differences in the metabolic and physiological state of the cells and the reduced complexity of the light regime during the MC addition experiment, as these cell cultures were continuously kept at constant light conditions and were thus not adapted to light-dark-cycles. Measuring the steady-state levels of the direct products of the reactions catalyzed by RubisCO reveal a dynamic behavior that follows the changes in light conditions. MC likely plays a role in that process as we know from previous studies that the *Microcystis* wild type and its MC-free mutant show significant metabolic differences under high light conditions, where MC-binding to RubisCO and other proteins is maximal [17].

We also show here that both the binding of MC to proteins and RubisCO condensate formation are reversible, since the intracellular free MC levels rise while its protein-binding fraction decreases, and RubisCO accumulations disappear from the membrane as the dark phase progresses and the next light phase approaches. Whether bound MC can be indeed removed from proteins, or the fraction of MC-free protein is increased by higher translational rates remains to be determined in further diel experiments. However, we know from previous studies that “microcystinilated” proteins are less susceptible to protease degradation and show enhanced stability [15], suggesting it is the reversibility of MC-binding rather than de novo synthesis of RubisCO and other proteins that is responsible for the observation of different degrees of MC bound to proteins in the diel cycle. The fact that we do not find a pronounced transcriptional response of neither RubisCO nor MC biosynthesis genes after 2 h of externally applied MC argues in favor of a rapid response to changing environmental conditions that avoids employing the gene expression machinery. Instead, the cell may utilize the transient nature of the predominantly non-covalent binding of MC to proteins to impart a higher degree of short-term metabolic regulation and flexibility.

The proposed reversible nature of MC-binding to proteins such as RubisCO introduce a new layer of metabolic flexibility that is likely beneficial to the multicellular lifestyle of *Microcystis*, especially within a dense bloom that has to deal with considerable fluctuations in the availability of light, CO_2_ and other resources [2,26]. Over the course of a day when photosynthesis and carbon fixation rates are high it is not uncommon for a *Microcystis* colony to face intervals of severe carbon limitations. As the CCM of some strains of *Microcystis* (including *M. aeruginosa* PCC 7806) lacks key importers of inorganic carbon [19], it may be advantageous to shift RubisCO from the carboxysome and place it in close proximity to other CCM and CBB components to form membrane-associated centers of carbon fixation. This concentrated clustering of carbon fixation enzymes outside of an enclosed biocompartment is reminiscent of the pyrenoid of some algal species that is described as a phase-separated liquid-like organelle [27]. Intriguingly, pyrenoids contain highly concentrated, spatially ordered, active RubisCO and they undergo rapid structural rearrangements in response to changes in carbon availability [28].

It remains to be determined how exactly the dynamic relocation of RubisCO and other CBB enzymes impacts the carbon fluxes and how that contributes to the remarkable persistence of *Microcystis* in the field. Our work, however, provides a next piece of information to help understand the biological function of MC and offers a good starting point for further experiments into that field.

## 4. Materials and Methods

### 4.1. Cultivation Conditions

*Microcystis aeruginosa* PCC 7806 wild type (WT) was cultivated in liquid BG-11 medium [29] at 23 °C under continuous illumination of 10 µmol photons m^−2^ s^−1^ without agitation. No external aeriation was used. This state of the culture was defined as low-light-adapted and will be referred to when mentioning growth under low light conditions. The growth was monitored by measuring the optical density at 750 nm.

For the light-dark-cycle experiments, four biological replicates of a low-light adapted *M. aeruginosa* WT culture were diluted with fresh BG-11 medium to an OD_750_ value of 0.1 and transferred to the diel conditions for adaptation. The setup of the light conditions was 16 h of daytime (55 µmol photons m^−2^ s^−1^) and 8 h of night (no illumination). The cultures were agitated mildly on an orbital shaker with 40 rpm without external aeriation at 25 °C in the AlgaeTron AG130 growth chamber (Photo Systems Instruments, Drasov, Czech Republic). The experiment was started during a dark phase when the cultures reached an OD_750_ of 0.25 (low cell density experiments) or 0.5 (high cell density experiments). Samples were taken each time the light conditions changed, as well as 2 h before and 2 h after that time point, resulting in 9 samples.

Addition experiments of MC to *M. aeruginosa* cultures were performed with low light-adapted cultures diluted with fresh BG-11 medium to an OD_750_ of 0.15 and incubated under low light conditions until an OD_750_ of 0.5 was reached. This starting culture was split in two, one aliquot had MC-LR (100 ng mL^−1^ final concentration) added to the liquid medium, the other an equal volume of the solvent (60% methanol). Cultures were incubated in a Multi-Cultivator MC 1000 (Photo Systems Instruments) with constant aeriation (ambient air) for 2 h. The experiment was carried out under high-light (250 µmol photons m^−2^ s^−1^) conditions in duplicates.

### 4.2. High Performance Liquid Chromatography (HPLC)

To analyze the intracellular peptides, cell pellets from 7.5 mL liquid culture were resuspended with 10 mL of 75% methanol and subsequently shaken for 5 min at 3200 rpm (Vortex Genie 2; Scientific Industries, Bohemia, NY, USA). After sonication for 10 min (70% amplitude, 3 s on/off pulse), the sample was centrifuged for 10 min (21,000× *g*, 10 min, 4 °C). The supernatant was transferred to a new reaction tube, and the pellet was resuspended with fresh 10 mL of 75% methanol. The extraction was repeated and both supernatants were pooled. To concentrate and purify the extracted peptides, the sample was diluted with water to a concentration of methanol of approx. 5% and run over a C-18 cartridge (Sep-Pak Plus C18 Cartridge; Waters, Milford, MA, USA). In the final step, the sample was eluted with 2 mL of 100% methanol and dried in a vacuum concentrator (RVC 2-25 CDplus; Christ, Osterode am Harz, Germany). For the analysis of extracellular peptides, supernatants of 7.5 mL *M. aeruginosa* liquid cultures were loaded directly onto C-18 cartridges and further processed like the pellet fraction.

The sample was resuspended with 200 µL of 60% methanol and filtered (Acrodisc 4 mm with 0.45 µm membrane; Pall Life Sciences, Port Washington, NY, USA) before 10–50 µL of it was loaded on the high-performance liquid chromatograph Prominence LC-20AD (Shimadzu, Kyoto, Japan) to analyze the peptides. The extracts were separated on a Symmetry Shield RP18 Column (100 Å, 3.5 µm, 4.6 mm × 100 mm) with a mobile phase containing 0.05% Trifluoroacetic acid. As a guard column a Symmetry Shield RP18 Sentry Guard Cartridge (100 Å, 3.5 µm, 3.9 mm × 20 mm) was used (both columns from Waters). The compounds were eluted at 1 mL min^−1^ using the following gradient of the 42 min program: (1) 70% aqua dest., 30% acetonitrile within 10 min to 65% aqua dest., 35% acetonitrile; (2) within 30 min to 30% aqua dest., 70% acetonitrile; (3) within 2 min to 100% acetonitrile. When only microcystin was examined, the program was shortened to 13 min using the following gradient: (1) 70% aqua dest., 30% acetonitrile within 12 min to 64% aqua dest., 36% acetonitrile; (2) within 1 min to 100% acetonitrile. The examination of the chromatograms and quantification of peaks was done with the LabSolutions software package (Version 5.87 SP1; Shimadzu, Kyoto, Japan). To collect certain compounds of the extract, like microcystin, the flow-through of the HPLC was collected with a fraction collector and dried in a vacuum concentrator to remove the acetonitrile. Afterward, the compound was resolved with 60% methanol. It was loaded on the HPLC to check if the fraction collection was performed successfully, and quantified, if necessary.

### 4.3. Protein Extraction

All the following steps were performed on ice or pre-cooled centrifuges. Cell pellets from 10 mL liquid culture were resuspended in 500 µL of native extraction buffer (50 mM Hepes; 5 mM MgCl_2_ × 6H_2_O; 25 mM CaCl_2_ × 2H_2_O; 10% glycerol; pH 7). Samples were sonicated on ice for 90 s (50% amplitude, 3 s/3 s on/off pulse, Sonopuls mini20, Bandelin, Berlin, Germany). Phenylmethylsulfonyl fluoride (PMSF) was added at a final concentration of 1 mM. Subsequently, residual unbroken cells were pelleted at 2000× *g*, 2 min, 4 °C and the supernatant was centrifuged at 21,000× *g* for 15 min, 4 °C, thus separating the cytoplasmic protein fraction (supernatant) from the membrane-associated protein fraction (pellet). To detach proteins from membranes, another round of sonication was performed. For that, the pellet was resuspended in 500 µL of native extraction buffer and sonicated for 60 s (50% amplitude, 3 s on/off pulse). Detached proteins were separated from membranes by centrifugation (21,000× *g* 15 min, 4 °C).

### 4.4. SDS-PAGE and Immunoblotting

For SDS-PAGE, the Bis-Tris buffer system was used (BiteSize Bio based on NuPAGE Invitrogen, Carlsbad, CA, USA). The used polyacrylamide concentration depended on the targeted protein, with higher polyacrylamide concentrations used for smaller proteins and vice versa (loading dye 5× concentrated; 250 mM Tris pH 6.8, 0.1% bromophenol blue, 50% glycerol, 10% SDS, 500 mM 2-mercaptoethanol). All protein samples were centrifuged for 1 min at 13,000× *g* to remove possible cell debris before loading on the gel. Protein concentrations were normalized to the optical density of the sampled liquid culture, resulting in equal amounts of protein loaded in each lane. The used protein ladder was PageRuler Plus Prestained (Thermo Fisher Scientific, Waltham, MA, USA). The gel was run at constant voltage of 180 V for 40 min in a MOPS running buffer (10.46 g L^−1^ MOPS, 6.06 g L^−1^ Tris, 1 g L^−1^ SDS, 0.3 g L^−1^ EDTA). Images were taken with the ChemiDoc XRS+ Imaging System (Bio-Rad, Feldkirchen, Germany).

For immunoblot analysis protein gels were blotted with a wet blot electrophoresis apparatus (Mini-Protean, Bio-Rad, Hercules, CA, USA) onto nitrocellulose membranes (Amersham Protein Premium 0.45 µm MC; GE Healthcare, Chicago, IL, USA) as described previously [30]. The transfer buffer (14.42 g L^−1^ Glycine, 3.03 g L^−1^ Tris) contained 20% methanol (*v*/*v*) for more efficient blotting. Membranes were blocked with 1% polyvinylpyrrolidone (PVP) K-30 in TBS-T (6.06 g L^−1^ Tris, 8.77 g L^−1^ NaCl, pH 7.4, 0.1% (*v*/*v*) Tween-20, Sigma-Aldrich, St. Louis, MO, USA) and washed once for 5 min, 4 °C with TBS-T. Primary antibodies were incubated in TBS-T overnight at 4 °C with the following concentrations: RbcL 1:10,000; RbcS 1:5000; CcmK 1:5000; MC 1:5000. Membranes were washed with TBS-T and secondary antibodies (α-mouse-IgG HRP-conjugate for MC and α-rabbit-IgG HRP-conjugate for all other antibodies) were applied in TBS-T and incubated for at least 1 h at 4 °C. Membranes were washed 4 times for 5 min, 4 °C, chemiluminescence was detected with the SERVALight Polaris CL HRP WB Substrate Kit (Serva, Heidelberg, Germany) and the ChemiDoc XRS+ Imaging System (Bio-Rad, Feldkirchen, Germany).

### 4.5. Immunofluorescence Microscopy (IFM)

Four milliliters of *M. aeruginosa* culture were pelleted for 1 min at 10,000× *g* and washed with 1 mL of PBS (8.18 g L^−1^ NaCl, 0.2 g L^−1^ KCl, 1.42 g L^−1^ Na_2_HPO_4_, 0.25 g L^−1^ KH_2_HPO_4_, pH 8.3). For fixation, the pellet was resuspended with 1 mL of 4% formaldehyde in PBS and incubated for 30 min at room temperature. After two washing steps with PBS the pellet was resuspended in 100 µL PBS, 20 µL were spread on a microscope slide, air-dried and stored at −20 °C for later use.

To start the hybridization with antibodies, the sample slides were equilibrated in PBS for 5 min at room temperature. Afterward, the slides were incubated with 2 mg mL^−1^ lysozyme in PBS-TX (PBS with 0.3% (*v/v*) Triton X-100, Sigma-Aldrich, Darmstadt, Germany) for 30 min at room temperature and washed twice with PBS-TX for 3 min. The samples were blocked with 1% PVP K-30 in PBS-T (PBS with 0.3% *v/v* Tween-20, Sigma-Aldrich, Darmstadt, Germany) for at least 1 h at 4 °C and washed twice with PBS-T. Primary antibody dilutions in PBS-T were as follows: RbcL 1:300 Rubisco large subunit form I, chicken (AS01 017 Agrisera, Vännas, Sweden); RbcS 1:200 (rabbit antiserum); CcmK 1:200 (rabbit antiserum); MC 1:250 (MC-LR MC10E7, mouse; Enzo Life Sciences, Lörrach, Germany). After incubation of at least 1 h at room temperature, the slides were washed twice with PBS-T and the secondary antibody (depending on primary antibodies used these were Alexa Fluor 488 goat anti-rabbit 1:200, Alexa Fluor 488 goat anti-mouse 1:100, Alexa Fluor 546 goat anti-chicken 1:200 and Alexa Fluor 568 goat anti-mouse 1:100; all from ThermoFisher Scientific. Waltham, MA, USA) was applied to the slides. Subsequently, the slides were washed twice, air-dried, embedded in approx. 30 µL of 4% propyl gallate in glycerol and covered with a coverslip. The slides were stored at −20 °C until use. Laser scanning confocal micrographs were taken with a Zeiss LSM 780 (Carl Zeiss, Oberkochen, Germany) using a Plan-Apochromat 63×/1.40 oil immersion objective. Alexa Fluor 488 was excited at 488 nm (detection spectrum 493–556 nm), Alexa Fluor 546 and 568 at 561 nm (570–632 nm), and autofluorescence at 633 nm (647–721 nm). The excitation was performed simultaneously.

### 4.6. Ultra-High Performance Liquid Chromatography Tandem Mass Spectrometry (UHPLC/MS) Sample Preparation and Measurement

To analyze and quantify metabolites of *M. aeruginosa* UHPLC was performed essentially as described previously [31]. The intracellular metabolites were extracted by resuspending pellets of 7.5 mL *M. aeruginosa* liquid cultures with 4 mL of H_2_O and subsequent sonication for 2 min (60% amplitude, 3 s on/off pulse). After centrifugation (21,000× *g*, 10 min, 4 °C), resulting supernatants were dried in a vacuum concentrator. For extracellular metabolite analysis, supernatants of liquid culture samples were filter-sterilized (Rotilabo-syringe filter 0.45 µm pore size; Carl Roth, Karlsruhe, Germany) and freeze-dried.

After resolving the dried extracts with 200 µL of H_2_O and filtration (0.2 µm filter Omnifix-F; B. Braun, Melsungen, Germany), the sample was analyzed by HPLC (LC-MS-8050 system; Shimadzu, Kyoto, Japan) and the incorporated LC-MS/MS method package for primary metabolites (version 2, Shimadzu, Kyoto, Japan). To prepare the extract to be loaded onto the system, 4 µL of extract was separated on a pentafluorophenylpropyl column (Supelco Discovery HS FS, Sigma Aldrich, Darmstadt, Germany; 3 µm, 150 × 2.1 mm) with a mobile phase containing 0.1% formic acid. Elution of the compounds was performed at 0.25 mL min^−1^ using the following gradient: 1 min 0.1% formic acid, 95% aqua dest., 5% acetonitrile, within 15 min linear gradient to 0.1% formic acid, 5% aqua dest., 95% acetonitrile followed by 10 min 0.1% formic acid, 5% aqua dest., 95% acetonitrile. Aliquots were continuously injected in the MS/MS part and ionized via electrospray ionization. Identification and quantification of the compounds was done with the LC-MS/MS method package and the LabSolutions software package (Shimadzu, Kyoto, Japan) using the multiple reaction monitoring values. Standard substances (Sigma-Aldrich, St. Louis, MO, USA) were included in all measurements and batches at varying concentrations for calibration.

### 4.7. Quantitative RT-PCR

Total RNA was extracted from a 10 mL culture sample using the TRIzol reagent (ThermoFisher Scientific, Waltham, MA, USA), purified by chloroform extraction, precipitated with isopropanol and resuspended in water. DNase digestion was performed on-column using the RNeasy Mini Kit 50 (Qiagen, Hilden, Germany). First strand cDNA synthesis with random primers was carried out with the Maxima reverse transcriptase (ThermoFisher Scientific, Waltham, MA, USA). Quantitative RT-PCR was carried out with the Blue S’Green qPCR mix (Biozym, Vienna, Austria) on a LightCycler 96 (Roche, Basel, Switzerland) at 2 step amplification conditions provided by the operating software. Reactions were performed in triplicates on 96 well plates. Specific primers were designed for the RNase P encoding housekeeping gene *rnpB* as a reference, for the RubisCO subunits *rbcL* and *rbcS*, for an NRPS gene from the microcystin biosynthetic gene cluster (*mcyA*) and for IPF47, a biosynthetic gene of a cryptic PKSI/PKSIII gene cluster [24]. Relative transcription was quantified with the Pfaffl method using the LinRegPCR software [32].

## Figures and Tables

**Figure 1 microorganisms-09-01265-f001:**
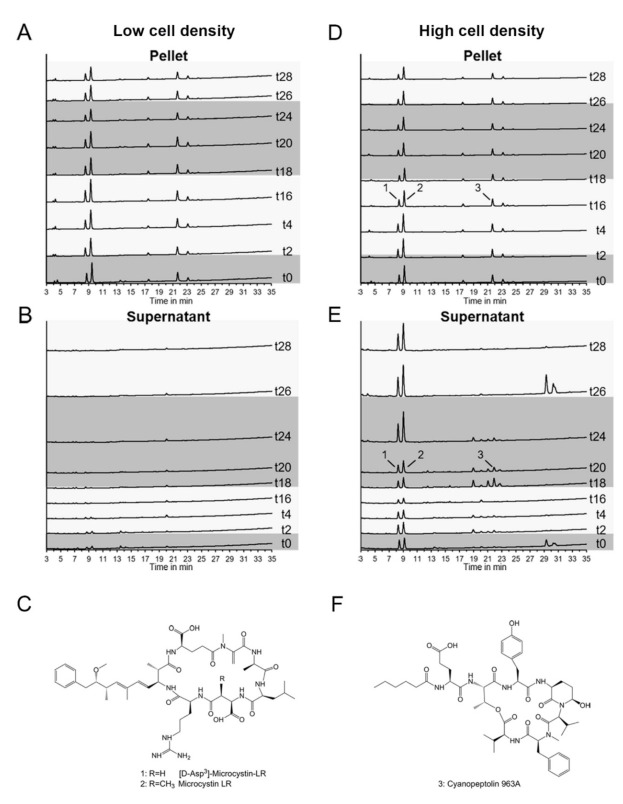
HPLC analysis of samples from a diel growth experiment of *Microcystis*. Cultures were grown at low (**A**,**B**) and high (**D**,**E**) cell densities, cell pellets or the culture supernatant were extracted at different time points (t0–t28, numbers indicate hours after the start of the experiment). Lighter and darker shading corresponds to light and dark phases, respectively. Two variants of microcystin (peaks 1 and 2, (**C**)) and one of cyanopeptolin (peak 3, (**F**)) were identified.

**Figure 2 microorganisms-09-01265-f002:**
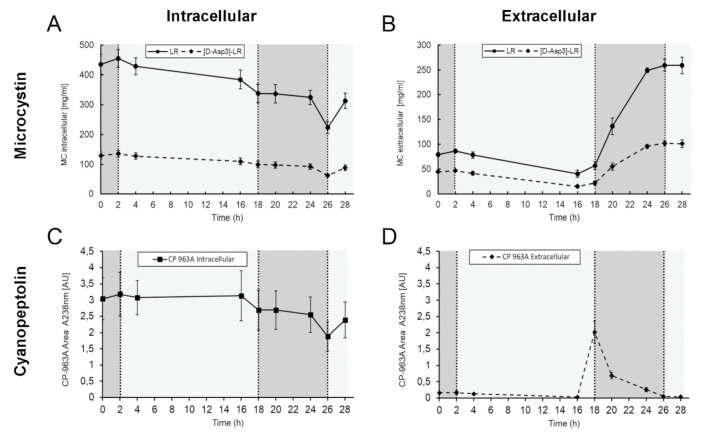
Quantification of microcystin and cyanopeptolin in samples from a diel growth experiment of *Microcystis*. Based on HPLC elution profiles of intracellular and extracellular extracts, two microcystin variants (**A**,**B**) and one of cyanopeptolin (**C**,**D**) were quantified with analytical standards. Lighter and darker shading corresponds to light and dark phases, respectively.

**Figure 3 microorganisms-09-01265-f003:**
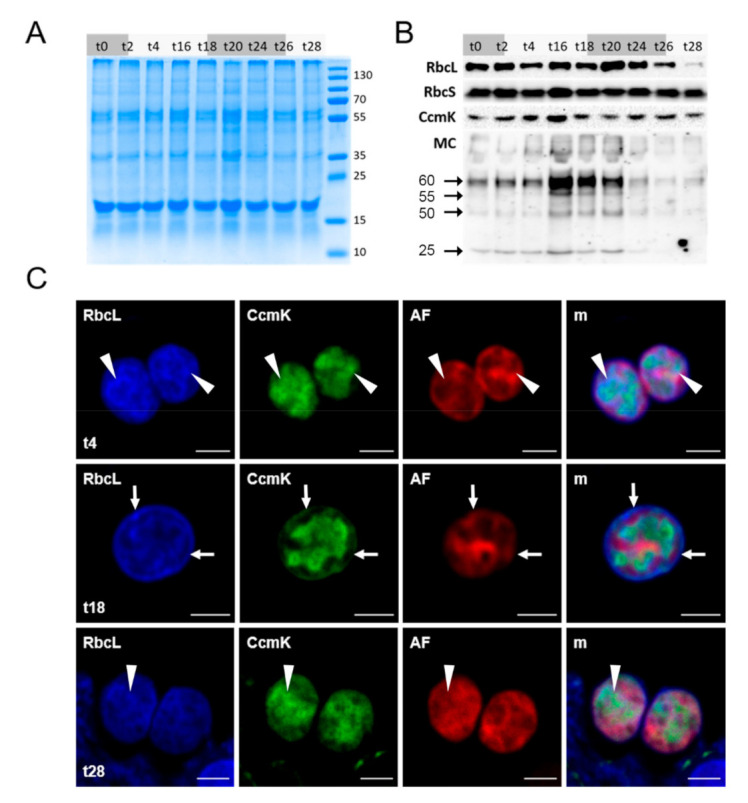
Fluctuations in microcystin-binding and localization of carbon fixation proteins in samples from a diel growth experiment of *Microcystis*. Protein samples (t0–t28, numbers representing hours since start of experiment) were separated by SDS-PAGE (**A**) and analyzed by immunoblots (**B**), detected proteins are named, molecular weight markers are given in kDa. MC binding to proteins increases during the light phase, is highest from t16 to t20 and decreases from t24 to t28. Cells were immunostained and analyzed by confocal microscopy (**C**), showing that RbcL transiently relocates to the cell membrane at t18. Arrows: regions of peripheral RbcL localization; arrowheads: cytosplasmic RbcL localization. RbcL/CcmK: primary antibodies used; AF: Chlorophyll *a* autofluorescence; m: merged fluorescence channels. Scale bars: 2 µm.

**Figure 4 microorganisms-09-01265-f004:**
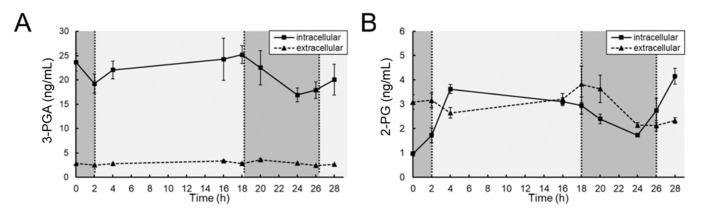
LCMS measurement of steady state levels of 3-PGA and 2-PG in samples from a diel growth experiment of *Microcystis*. The two products of the carboxylase and oxygenase activities of RubisCO, 3-phosphoglycerate (3-PGA, (**A**)) and 2-phosphoglycolate (2-PG, (**B**)), respectively, were quantified intra-and extracellularly.

**Figure 5 microorganisms-09-01265-f005:**
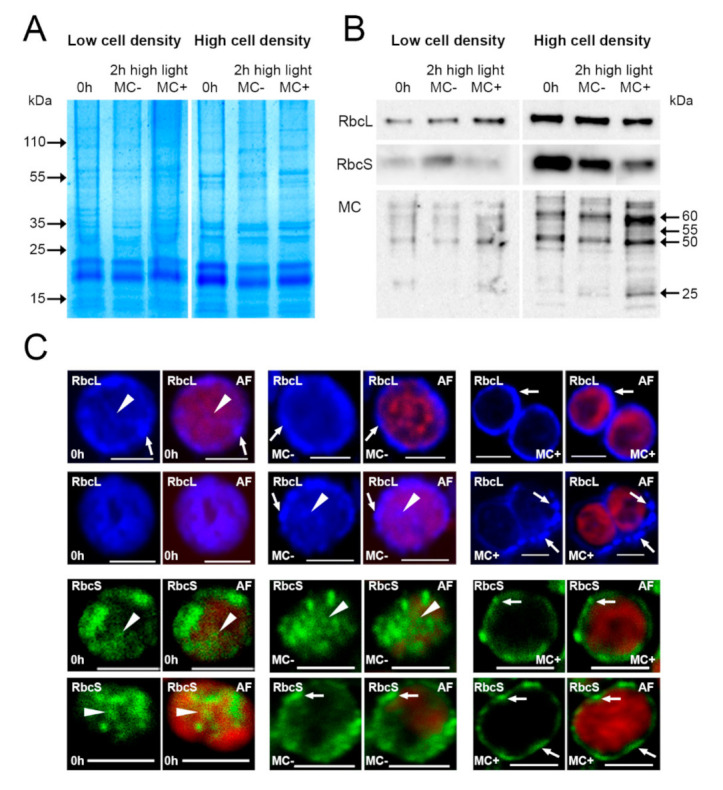
Influence of microcystin added to the growth medium of a liquid culture of *Microcystis*. Liquid cultures at low and high cell densities were grown at high light with the addition of MC to the growth medium (MC+) or without it (MC−). Protein extracts were analyzed by SDS-PAGE (**A**) and immunoblots (**B**), detected proteins are named, molecular weight markers are given in kDa. MC binding to proteins increases after the addition of MC. Cells from high cell density cultures were immunostained and analyzed by confocal microscopy (**C**) showing a relocation of both RbcL and RbcS to the cell membrane only after MC addition. RbcL/RbcS: primary antibodies used; AF: Chlorophyll *a* autofluorescence in red fluorescence channel. Distinct accumulations of RbcL and RbcS are highlighted by arrowheads (cytoplasmic) and arrows (peripheral). Scale bars: 2µm.

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
