# Peer review of "Diel Variations of Extracellular Microcystin Influence the Subcellular Dynamics of RubisCO in Microcystis aeruginosa PCC 7806"

_microorganisms, 2021, doi:10.3390/microorganisms9061265_

Round 1

Reviewer 1 Report

the manuscript is nice and novel, it deserves publication, I just have a remark concerning the use of the adjective 'diurnal' for the experiment carried out, this and other small notes are in the uploaded pdf

Author Response

Response to Reviewer 1 Comments:

Thank you very much for the thorough and critical reading of our manuscript, your comments have been very helpful. We have addressed your points as follows:

Point 1: I just have a remark concerning the use of the adjective 'diurnal' for the experiment carried out, this and other small notes are in the uploaded pdf

Response 1: We have exchanged the word "diurnal" in the revised manuscript, mostly for "diel" but also for "light-dark-cycle" where it refers to the experiment in question. We have also changed the title accordingly.

As to the comments added to the text:

  • regarding the connection of high light and low carbon (mentioned in the abstract), it is understood that within cyanobacterial blooms the carbon availability drops locally as a result of increased light intensities and thus accelerated photosynthesis. We address the issue in the discussion, we do not feel that this needs to be elaborated in the abstract
  • we have corrected the use of "cytosol/cytosolic" to "cytoplasma/cytoplasmic", we now also use "cell membrane" rather than "cytoplasmic membrane"
  • we have also trimmed and stream-lined the methods section, especially in the description of the design for the light-dark-cycle experiment and the sampling scheme. We feel now that a graphic representation should not be necessary, instead, in our figures we employ a system of shaded boxes to indicate light and dark phases and time points of sampling
  • we have abandoned the use of "day" when referring to the light phase of the light-dark-cycle experiment and have made the terminology consistent throughout the revised manuscript
  • we have now restricted the use of the word "dynamics" to fewer, more appropriate instances; we use "fluctuations" and "variations" now more frequently
  • the figure legends have been updated and include now descriptions of the displayed data; however, we feel that some methodological information is necessary for understanding and have kept it to some degree
  • we have addressed all other minor comments regarding spelling, grammar, wording and style

Reviewer 2 Report

  1. In Figure 5C, the difference in spatial distribution pattern of RbcL between MC- and MC+group (MC feeding experiment) is significant, as shown in the Figure 5C, the RbcL shifted to be around the cytoplasma membrane region once MC are added.  This is very interesting. However, in Figure 3C, compared to time point t4, the significantly higher proportion of cytoplasmic membrane-associated RbcL was observed at the time point t18 where the extracellular microcystin level is almost similar to the time point t4. This observation appear to negate the association of spatial distribution of RbcL with extracellular MC. This needs to be addressed in the discussion.
  2. Please double check the protein marker in the blotting figure, Zillliges et al previously showed the ~55KD of major MC-binding proteins, however, the blotting picture of this work indicated the higher, 60KD of major MC-binding ones. 
  3. In the first line of the abstract, Microcystis should be italicized.
  4. In the 5th line of discussion section, the format of words "evidence have accumulated showing" should be corrected.

Author Response

Response to Reviewer 2 Comments:

Thank you very much for the thorough and critical reading of our manuscript, your comments have been very helpful. We have addressed your points as follows:

Point 1: In Figure 5C, the difference in spatial distribution pattern of RbcL between MC- and MC+group (MC feeding experiment) is significant, as shown in the Figure 5C, the RbcL shifted to be around the cytoplasma membrane region once MC are added.  This is very interesting. However, in Figure 3C, compared to time point t4, the significantly higher proportion of cytoplasmic membrane-associated RbcL was observed at the time point t18 where the extracellular microcystin level is almost similar to the time point t4. This observation appear to negate the association of spatial distribution of RbcL with extracellular MC. This needs to be addressed in the discussion.

Response 1: We do not claim a direct influence of the externally applied MC on the relocation of RbcL in the text, instead, the connecting parameter here seems to be the "microcystinilation" of intracellular proteins that coincides (or perhaps directly precedes) the relocation. This is true for both experiments, the light-dark-cycle experiment and the MC addition experiment. In the light-dark experiment the intracellular proteins are microcystinilated first, leading to a relocation of Rubisco and then MC appears in the external medium. In the addition experiment, the external MC is the first to appear (because we added it artificially), the cell seems to perceive this signal and as a result the microcystinilation of proteins is initiated. As before, this leads to a relocation of Rubisco. The slight difference in the order of events could be explained by the different experimental designs and thus by the different metabolic and physiological states the cells are in before and during the experiment. We have added a section in the discussion that clarifies this issue.

Point 2: Please double check the protein marker in the blotting figure, Zillliges et al previously showed the ~55KD of major MC-binding proteins, however, the blotting picture of this work indicated the higher, 60KD of major MC-binding ones. 

Response 2: The marker is correct as displayed, we frequently observe variations in the band pattern of MC binding in Western blots. The 55kD and 60kD bands are not always the predominant ones. But this is a whole complex topic in itself we could not fully address in this publication.

Points 2 and 3: In the first line of the abstract, Microcystis should be italicized. In the 5th line of discussion section, the format of words "evidence have accumulated showing" should be corrected.

Responses 2 and 3: We have corrected these issues in the revised manuscript.

Reviewer 3 Report

Very nice work, well written and it provides interesting data on the description of the biological roles of microcystins for the organisms which synthesize them. Just minor comments need to be reviewed before publication.

  • Fig. 1: Peaks 1, 2 and 3 should be specified.
  • References indicated in superscript should be put before the point and not after (check especially for the introduction section). 

Author Response

Response to Reviewer 3 Comments:

Thank you very much for the thorough and critical reading of our manuscript, your comments have been very helpful. We have addressed your points as follows:

Point 1: Fig. 1: Peaks 1, 2 and 3 should be specified.

Response 1: The chemical structures corresponding to the three peaks from the chromatograms are given in Fig. 1C and 1F, the figure legend has been updated and now specifies which substance is assigned to which peak.

Point 2: References indicated in superscript should be put before the point and not after (check especially for the introduction section).

Response 2: We have corrected this in the revised manuscript.